# Fabrication of Liquid Scintillators Loaded with 6-Phenylhexanoic Acid-Modified ZrO_2_ Nanoparticles for Observation of Neutrinoless Double Beta Decay

**DOI:** 10.3390/nano11051124

**Published:** 2021-04-27

**Authors:** Akito Watanabe, Arisa Magi, Akira Yoko, Gimyeong Seong, Takaaki Tomai, Tadafumi Adschiri, Yamato Hayashi, Masanori Koshimizu, Yutaka Fujimoto, Keisuke Asai

**Affiliations:** 1Department of Applied Chemistry, Graduate School of Engineering, Tohoku University, 6-6-07 Aoba, Aramaki, Aoba-ku, Sendai 980-8579, Japan; arsm723.tmsj@gmail.com (A.M.); hayashi@aim.che.tohoku.ac.jp (Y.H.); koshi@qpc.che.tohoku.ac.jp (M.K.); fuji-you@qpc.che.tohoku.ac.jp (Y.F.); asai@qpc.che.tohoku.ac.jp (K.A.); 2WPI-AIMR, Tohoku University, 2-1-1 Katahira, Aoba-ku, Sendai 980-8577, Japan; akira.yoko.c7@tohoku.ac.jp (A.Y.); tadafumi.ajiri.b1@tohoku.ac.jp (T.A.); 3New Industry Creation Hatchery Center, Tohoku University, 6-6-10 Aoba, Aramaki, Aoba-ku, Sendai 980-8579, Japan; kimei.sei.c6@tohoku.ac.jp; 4Institute of Multidisciplinary Research for Advanced Materials, Tohoku University, 2-1-1 Katahira, Aoba-ku, Sendai 980-8577, Japan; takaaki.tomai.e6@tohoku.ac.jp

**Keywords:** ZrO_2_, liquid scintillator, neutrinoless double beta decay, hydrothermal synthesis, nanoparticles, 6-phenylhexanoic acid

## Abstract

The observation of neutrinoless double beta decay is an important issue in nuclear and particle physics. The development of organic liquid scintillators with high transparency and a high concentration of the target isotope would be very useful for neutrinoless double beta decay experiments. Therefore, we propose a liquid scintillator loaded with metal oxide nanoparticles containing the target isotope. In this work, 6-phenylhexanoic acid-modified ZrO_2_ nanoparticles, which contain ^96^Zr as the target isotope, were synthesized under sub/supercritical hydrothermal conditions. The effects of the synthesis temperature on the formation and surface modification of the nanoparticles were investigated. Performing the synthesis at 250 and 300 °C resulted in the formation of nanoparticles with smaller particle sizes and higher surface modification densities than those prepared at 350 and 400 °C. The highest modification density (3.1 ± 0.2 molecules/nm^2^) and Zr concentration of (0.33 ± 0.04 wt.%) were obtained at 300 °C. The surface-modified ZrO_2_ nanoparticles were dispersed in a toluene-based liquid scintillator. The liquid scintillator was transparent to the scintillation wavelength, and a clear scintillation peak was confirmed by X-ray-induced radioluminescence spectroscopy. In conclusion, 6-phenylhexanoic acid-modified ZrO_2_ nanoparticles synthesized at 300 °C are suitable for loading in liquid scintillators.

## 1. Introduction

The identification of neutrinoless double beta decay (0*νββ*) events has become important toward understanding neutrino properties ever since neutrino mass was confirmed through the observation of neutrino oscillation [1,2]. 0*νββ* is a radioactive event in which an even–even nucleus transforms into a lighter isobar containing two more protons with the emission of only two electrons. The occurrence of 0*νββ* would confirm that a neutrino has finite mass and is a Majorana particle, which means that a neutrino is its own antiparticle. However, the detection of 0*νββ* is challenging, because the estimated half-lives of the candidate isotopes for the decay mode are extremely long, typically more than 10^25^ years [3]. The performances of 0*νββ* detectors are parameterized with the experimental sensitivity, $F_{D}^{0\nu }$ [4]. The experimental sensitivity is shown below:(1)$$F_{D}^{0\nu }=\ln2\frac{\eta \epsilon N_{A}}{A}\sqrt{\frac{T_{m}M}{B\Delta }}$$
where η is the isotopic abundance of the candidate isotope, ϵ is the detection efficiency, $N_{A}$ is the Avogadro’s number, A is the mass number, $T_{m}$ is the measurement time, M is the total detector mass, B is the background level, and Δ is the energy resolution. Therefore, to detect 0*νββ*, which has an extremely low number of events, a large and efficient detector containing tens or hundreds of kilograms of candidate isotopes is required. Therefore, the measurement of 0*νββ* requires a large and efficient detector containing tens or hundreds of kilograms of the candidate isotopes for detection. 0*νββ* can be distinguished from ordinary double beta decay with two neutrino emissions based on the difference in the total energy of the two beta rays, which requires detectors with high energy resolution. To achieve these specifications, various types of detector systems, such as bolometers, Ge semiconductor detectors, and liquid scintillators, have been proposed [3].

Candidate isotope-loaded liquid scintillators are one of the detector systems useful for 0*νββ* experiments. A liquid scintillator, which is mainly composed of an organic solvent and phosphor, is a liquid that converts ionizing radiation into visible photons. The advantage of liquid scintillators is that large detectors with high uniformity can be constructed at a low cost; in addition, limited energy resolution can be sufficient. However, to accumulate adequate detection events to provide sufficient statistics in a reasonable time period, the liquid scintillator must contain the candidate isotope at concentrations in the range of 0.1–10 wt.% [5] without severe degradation of the optical properties of the liquid scintillator. Several research groups have developed metal-loaded liquid scintillators by dissolving organometallic molecules that contain the candidate isotope [5,6,7,8]; however, the excited state can be quenched by these incorporated molecules, resulting in the reduction of the scintillation light yield. Moreover, the concentration of candidate isotopes is limited by the solubility of the metal compound and some of the incorporated molecules may be degraded via photo-induced oxidation in the presence of oxygen [5]. 

As an alternative approach, we propose liquid scintillators loaded with metal oxide nanoparticles that contain the candidate isotopes for 0*νββ*. In this approach, we expect that a large amount of the candidate isotope can be loaded into the liquid scintillator, because a dispersion of nanoparticles in an organic solvent is reportedly stable for a long period at remarkably high concentrations (up to 77 wt.%) [9,10]. Nanoparticles with diameters less than one-tenth of the wavelength of light can reduce Rayleigh scattering. Therefore, liquid scintillators with optical transparency can be fabricated upon the addition of nanoparticles. Surface modification of the nanoparticles is effective for dispersing inorganic nanoparticles in hydrophobic solutions used for liquid scintillators. It has been reported that the aggregation of nanoparticles can be suppressed and that the nanoparticles can be dispersed in an organic solvent by tuning the affinity of the nanoparticle surface upon surface modification using a hydrophobic organic solvent [10,11,12,13].

Sub/supercritical hydrothermal methods are suitable for the synthesis of such nanoparticles whose surface is modified by organic molecules. These methods provide unique reaction conditions for nanoparticle formation, allowing in situ modification of their surface. The dielectric constant of water decreases upon increasing the temperature, slowly approaching that of organic compounds. As the temperature of the aqueous solution increases, the solubility of the metal oxide decreases causing a high degree of supersaturation and the formation of nanoparticles [14,15,16,17]. Organic molecules are miscible with water at high temperatures under sub/supercritical conditions, which allows the surface modification of nanoparticles [18,19,20]. To date, many studies have been conducted toward the sub/supercritical hydrothermal synthesis of surface-modified metal oxide nanoparticles [20,21,22,23].

The solvents used in liquid scintillators are generally benzene derivatives, particularly alkyl benzenes, such as toluene, xylene, and cumene. Although many studies have been reported on long-chain carboxylic acid-modified nanoparticles, which are highly dispersible in saturated aliphatic organic solvents such as cyclohexane, their dispersibility in toluene is very low with a stable dispersion concentration of only 0.01 wt.% [9,23]. Therefore, our group proposed a method of modifying the surface of nanoparticles using aromatic carboxylic acids in which a benzene ring is introduced at the tip of a linear carboxylic acid to improve the dispersibility in aromatic organic solvents. Using 3-phenylpropionic acid or 6-phenylcaproic acid as a modifier, we successfully dispersed the nanoparticles in toluene [24,25]. 

In this study, we focused on the synthesis of organically modified ZrO_2_ nanoparticles, which contain ^96^Zr as a candidate isotope for 0*νββ*. Since the double beta decay of ^96^Zr has a high *Q* value of 3350 keV [8], it is less susceptible to gamma-ray environmental background and a clear 0*νββ* peak in the energy spectrum is expected. Since we have previously shown that the modification of nanoparticles improves their dispersibility in toluene [24], the sub/supercritical hydrothermal synthesis of ZrO_2_ nanoparticles was performed using 6-phenylhexanoic acid. The effect of the synthesis temperature on the formation and surface modification of the nanoparticles is discussed. To develop a scintillator containing a large amount of the candidate isotope, the dispersion was used as the liquid scintillator. It has been reported that adding a large number of nanoparticles to organic scintillators results in increased light scattering and reduces the scintillation light yield [26]. Therefore, X-ray-induced radioluminescence spectroscopy was used to determine whether the scintillation peak of the fabricated scintillator was present.

## 2. Experiment 

ZrO_2_ nanoparticles were synthesized using subcritical/supercritical hydrothermal methods [14,15,16,17] in batch reactors (inner volume, 5 mL; AKICO Corp., Tokyo, Japan) made of Hastelloy C, which is a nickel-based alloy with high resistance to corrosion at high temperature. The precursor solution consisting of ZrO_2_ nanoparticles was prepared by dissolving ZrOCl_2_∙H_2_O (99.0%, FUJIFILM Wako Pure Chemical Corp., Osaka, Japan) in distilled water at a concentration of 0.1 mol/L. The pH was adjusted to 5.8, which is greater than the p*K*a of 6-phenylhexanoic acid (=4.78), by adding KOH (FUJIFILM Wako Pure Chemical Corp., Osaka, Japan), following the procedure in previous studies [24,25]. This precursor solution (4.12, 3.75, 3.22, or 1.75 mL) was added to the batch reactors for synthesis at 250, 300, 350, or 400 °C, respectively. These volumes correspond to a final pressure of 30 MPa at the specified reaction temperatures. As an organic modifier, 6-phenylhexanoic acid (PHA; Tokyo Chemical Industry Co., LTD., Tokyo, Japan) was added to the precursor solution such that the molar ratio of PHA to Zr was 6:1. The reactions were performed at each temperature for 10 min while shaking and then cooled to room temperature in water. The reactors were rinsed several times using 3.75 mL of toluene (99.5%; FUJIFILM Wako Pure Chemical Corp., Osaka, Japan) and water to collect the products. The collected organic and aqueous phases were separated upon standing overnight. The organic phase was recovered and used for further experiments. 

To obtain the powdered nanoparticles, ethanol (99.5%; FUJIFILM Wako Pure Chemical Corp., Osaka, Japan) was added to the organic phase, the solution was subjected to centrifugation, and the supernatant was removed to separate the unreacted chemicals. This procedure was repeated three times. The remaining particles were dried on a petri dish at room temperature and used for further characterization.

To determine the crystallographic phases of the nanoparticles, the samples were analyzed using X-ray diffraction (XRD; Ultima IV, Rigaku Corp., Tokyo, Japan) [27] with Cu Kα radiation in a 2*θ*–*θ* set-up; the scan interval and rate were 0.02° and 4.0°/min, respectively. The crystallite sizes were estimated using the Halder–Wagner method [28] via Rietveld fitting with the RIETAN-FP [29] code. The Halder–Wagner equation is shown below:(2)$${\left(\frac{\beta }{\tan\;\theta }\right)}^{2}\;=\;\frac{K\lambda }{D}\;\times \;\frac{\beta }{\tan\;\theta \;\sin\;\theta }\;+\;16\varepsilon^{2}$$
where *β* is the full width at half maximum of the diffraction peak (in radians), *θ* is the Bragg angle, *K* is the shape factor for the mean volume-weighted size of spherical crystallites (4/3) [30], *λ* is the wavelength of the X-ray (0.154184 nm), *D* is the crystallite size, and *ε* is the microstrain of the crystal. The shapes and sizes of the nanoparticles were observed using transmission electron microscopy (TEM; HD-2700, Hitachi High-Technologies Corp, Tokyo, Japan) operated at 200 kV [31]. The nanoparticle diameters were obtained as the average diameter of 50 particles with error bars representing the standard deviation. Fourier-transform infrared (FTIR) spectroscopy was carried out on an FTIR spectrometer (Nicolet 6700, Thermo Fisher Scientific K. K., Tokyo, Japan) [32]. Measurements were conducted in the range of 4000–650 cm^−1^, which is the measurable wavenumber of the device. The thermal properties of the nanoparticle powders were measured by thermogravimetric analysis (TGA; SDT Q600, TA Instruments Japan Inc., Tokyo, Japan) [33] in the temperature range from room temperature to 600 °C at a heating rate of 10 °C/min under a flow of air at 100 mL/min. 

Dynamic light scattering (DLS; Nano-ZS, Malvern Panalytical Ltd., Malvern, UK) measurements [34] were performed to investigate the aggregated state of the nanoparticles in the toluene dispersion. An inductively coupled plasma atomic emission spectroscopy (ICP-AES) analysis [35] was performed (iCAP6500, Thermo Fisher Scientific K. K., Tokyo, Japan) at an emission wavelength of 339 nm to determine the Zr concentration in the nanoparticle dispersion. The absorption spectra of the dispersions were measured using a spectrophotometer (U-3500, Hitachi Ltd., Tokyo, Japan). The measurement was performed using a quartz cell with a 1-mm optical length.

Nanoparticle dispersions were prepared by concentrating the organic phase containing dispersed particles. Concentration was carried out using 15 mL of toluene dispersion obtained by collecting the organic phase. The dispersions were placed in screw vials, in which the spouts were covered with a perforated aluminum foil and stored at 50 °C to evaporate the solvent until precipitation occurred, which results in the maximum soluble state of the nanoparticles in toluene [10,36]. After the precipitation occurred, the supernatants were extracted to obtain nanoparticle dispersions of about 5 mL.

The liquid scintillator was fabricated using a concentrated nanoparticle dispersion. 2,5-Diphenyloxazole (DPO; Dojindo Laboratories, Kumamoto, Japan) and 1,4-bis(5-phenyl-2-oxazolyl)benzene (POPOP; Dojindo Laboratories, Kumamoto, Japan) were used as phosphors in the liquid scintillator. DPO and POPOP were first mixed homogeneously in a weight ratio of 80:1, and the mixed powder was then added to the toluene dispersion at a concentration of 100 g/L. The X-ray-induced radioluminescence spectra of the liquid scintillator loaded with ZrO_2_ nanoparticles was measured with the same measurement system as in the previous study [37]. The liquid scintillator was contained in a quartz cell and irradiated with X-rays from an X-ray generator (D2300-HK, Rigaku Corp., Tokyo, Japan) equipped with a Cu target operated at 40 kV and 40 mA. A charge-coupled device (CCD)-based spectrometer (QE Pro Spectrometer, Ocean Insight Inc., Tokyo, Japan) was used to record the radioluminescence spectra.

## 3. Results and Discussion

Figure 1 shows the XRD patterns of the ZrO_2_ nanoparticles synthesized at different temperatures. The XRD patterns of the nanoparticles synthesized at 250 and 300 °C can be attributed to the tetragonal ZrO_2_ phase, whereas those synthesized at 350 and 400 °C were attributed to both tetragonal and monoclinic phases. The diffraction peaks of the XRD patterns of the nanoparticles synthesized at 250 and 300 °C were broader than those synthesized at 350 and 400 °C. These results indicate that the crystallite size decreased when the synthesis was performed at 250 and 300 °C. The crystallite diameters were calculated by fitting using Equation (2). Figure 2 shows Halder-Wanger plots of ZrO_2_ nanoparticles synthesized at different temperatures. The crystallite sizes of the nanoparticles synthesized at 250, 300, 350, and 400 °C were 1.5 ± 0.1, 1.8 ± 0.1, 4.4 ± 2.2, and 5.4 ± 2.4 nm, respectively, suggesting that single-nanometer-sized crystallites were synthesized under all the temperature conditions studied.

Figure 3 shows the TEM images of the ZrO_2_ nanoparticles synthesized at different temperatures. Spherical nanoparticles were formed under all the reaction temperatures investigated. The diameters of the nanoparticles synthesized at 250, 300, 350, and 400 °C were 4.0 ± 1.3, 3.7 ± 1.2, 6.7 ± 1.8, and 5.6 ± 1.4 nm, respectively. The particle sizes of the nanoparticles synthesized at 250 and 300 °C were smaller than those synthesized at 350 and 400 °C. The particle sizes formed at different reaction temperatures showed a similar trend as the corresponding crystallite sizes estimated from the XRD patterns. In addition, the aggregation of nanoparticles was observed in the TEM images of all the samples.

The chemical states of the surface-modifying molecules on the nanoparticles were evaluated using FTIR spectroscopy, and the results are shown in Figure 4. All the ZrO_2_ nanoparticles exhibit bands at 650–850 cm^−1^ assigned to the Zr−O modes of the ZrO_2_ nanoparticles [38], in addition to weak bands at 2950 and 2820 cm^−1^ assigned to the asymmetric and symmetric stretching modes of −CH_2_− in the alkyl chain of PHA, respectively [39,40]. Two strong peaks at 1540 and 1410 cm^−1^ were also observed, which were assigned to the asymmetric and symmetric stretching modes of the carboxylate group (−COO^−^) in PHA, respectively [38,41]. The presence of these bands indicates that the surface-modifying molecules are attached to the surface of the ZrO_2_ nanoparticles via coordination bonds to the carboxyl group [39,40,42]. In addition, no peak was detected at ~1708 cm^−1^, which corresponds to the stretching mode of the −COOH group. These results indicate that the nanoparticles were almost free from unreacted organic surface modifier on their surface. The strongest peak in the spectra was observed for the nanoparticles synthesized at 300 °C, which corresponds to the carboxylate group, while the sample synthesized at 250 °C exhibited the next strongest peak. The spectra obtained for the samples synthesized at 350 and 400 °C showed relatively weaker peaks. These results indicate that the number of organic surface modifier on the surface of the nanoparticles synthesized at 250 and 300 °C is larger than those synthesized at 350 and 400 °C.

TGA was performed to quantitatively analyze the number of surface-modified molecules present on the nanoparticles, and the TGA curves obtained for the nanoparticles are presented in Figure 5. All the TGA curves showed a significant weight loss at >250 °C, which corresponds to the weight of the surface modifiers attached to the surface of the nanoparticles [39,40]. The weight loss assigned to the desorption of the surface modifiers ended near 500 °C for any of the samples. The surface organic modification density of the ZrO_2_ nanoparticles is estimated from the weight loss observed from 250 to 500 °C and presented in Table 1. These results were consistent with those obtained using FTIR spectroscopy. As the synthesis temperature increased from 250 to 300 °C, the modification density of the nanoparticles increased from 2.8 ± 0.0 to 3.1 ± 0.2 molecules/nm^2^. On the other hand, the modification densities of the nanoparticles synthesized at 350 and 400 °C were <1 molecules/nm^2^. Consequently, the modification density decreases as the synthetic conditions approach the supercritical state, which was similar to the results obtained for HfO_2_ nanoparticles in a previous study [43]. The difference in the affinity between the crystalline phase and modifier molecules appears to be related to the decrease in the modification density. Since the surface of tetragonal ZrO_2_ has more than three times the number of basic sites on the surface of monoclinic ZrO_2_ [44], its reactivity with 6-phenylhexanoic acid is considered to be higher. Therefore, the nanoparticles synthesized at 350 and 400 °C, which are rich in monoclinic phases, may reduce the modification density.

Figure 6 shows the photographic images obtained for the concentrated nanoparticle dispersions prepared using samples synthesized at different temperatures. The dispersions were stable for at least one month, and no precipitation was observed. The dispersions synthesized at 250–350 °C showed pale turbidity, while those synthesized at 400 °C were transparent. When the dispersions were irradiated with green laser light to observe the Tyndall scattering, significant scattering was observed in the dispersions synthesized at 250–350 °C but not at 400 °C, which was strongly related to the concentration of ZrO_2_ nanoparticles.

The number size distributions of the ZrO_2_ nanoparticles in the toluene dispersions measured using DLS are shown in Figure 7. The particle sizes of the nanoparticles in the dispersions synthesized at 250, 300, 350, and 400 °C were estimated to be 38.7 ± 6.6, 26.5 ± 5.5, 26.9 ± 5.9, and 31.9 ± 7.7 nm, respectively. When compared with the crystallite sizes estimated from the XRD patterns and the particle sizes estimated using TEM, DLS indicated the presence of significantly larger particles, which indicates that the nanoparticles were aggregated in the dispersions. The aggregate size was homogeneous and maintained a dispersive state without further agglomeration or phase separation.

The Zr concentrations in the nanoparticle dispersions are summarized in Table 1. The highest Zr concentration of 0.33 ± 0.04 wt.% was obtained for the nanoparticles synthesized at 300 °C. The Zr concentrations of the dispersions obtained from the particles synthesized at 250 and 300 °C were an order of magnitude higher than those of the particles synthesized at 350 and 400 °C. The dispersibility and solubility of these nanoparticles can be improved by increasing the surface modification density [9,10,36]. Table 1 shows that the surface modification densities of nanoparticles synthesized at 250 and 300 °C were higher than those synthesized at 350 and 400 °C. These results demonstrate the correlation between the dispersed ZrO_2_ concentration and surface modification density.

The liquid scintillators were fabricated using the ZrO_2_ nanoparticle dispersions. Figure 8 shows the absorption spectra of the toluene dispersion of ZrO_2_ nanoparticles. In all dispersions, the absorbance below 500 nm was enhanced upon loading the ZrO_2_ nanoparticles in toluene. This increase in the absorbance was attributed to Rayleigh scattering by the aggregated nanoparticles, which was observed using DLS. The absorbance in this range was the highest in the dispersion of particles synthesized at 350 °C, followed by those at 250 and 300 °C. The bandgap of tetragonal and monoclinic ZrO_2_ nanoparticles are 3.6 and 4.5 eV, respectively, according to a previous paper [45], and the wavelengths corresponding to the bandgap energies are 344 and 276 nm, respectively. Since the ZrO_2_ nanoparticles synthesized at 350 °C contain a monoclinic phase, the absorbance increased significantly in the range of 300 to 350 nm. In addition, it is considered that the dispersion of particles synthesized at 400 °C has a low absorbance, because the Zr concentration was small, which also correlates with the results shown in Figure 6. The absorbance was <0.1 for wavelengths >410 nm, which suggests that the ZrO_2_ nanoparticle dispersions have high transparency in the region of POPOP emission. The liquid scintillators were prepared by dissolving the DPO and POPOP phosphors in these dispersions. Figure 9 shows the X-ray-induced radioluminescence spectra obtained for the liquid scintillators. The energy of the X-ray is 8.048 keV. In all samples, a dominant band attributed to the emission from POPOP is observed at 425 nm. No significant decrease in the peak intensity of emission was observed in any of the samples. This result indicates that the energy transfer from toluene to POPOP via DPO occurs successfully even in the liquid scintillator, which contains 0.33 ± 0.04 wt.% of Zr. A similar scintillation spectrum is expected for high-energy β-rays, because the scintillation was caused by high-energy electrons produced via the photoelectric effect. Considering the absorption spectra shown in Figure 8, the liquid scintillator is transparent to the scintillation wavelength.

## 4. Conclusions and Outlook

6-phenylhexanoic acid-modified ZrO_2_ nanoparticles were synthesized to assess their suitability for use in double beta decay experiments. The ZrO_2_ nanoparticles synthesized at lower temperatures (250 and 300 °C) have smaller particle sizes and larger surface modification densities than those synthesized at higher temperatures (350 and 400 °C). The highest modification density of 3.1 ± 0.2 molecules/nm^2^ was obtained at 300 °C, which resulted in the highest concentration of dispersed nanoparticles in toluene (0.33 ± 0.04 wt.%-Zr). The ZrO_2_ nanoparticle-loaded liquid scintillator was transparent to the scintillation wavelength, and an apparent scintillation peak from the phosphor emission was observed at 425 nm in the X-ray-induced radioluminescence spectra. As a result, a toluene-based liquid scintillator with a relatively high concentration of ZrO_2_ nanoparticles was successfully fabricated. 

Currently, the maximum dispersed concentration of Zr was 0.33 wt%. With a natural isotope ratio of Zr, 9.24 × 10^−3^ wt.% of ^96^Zr is present in the liquid scintillator. Even for a 100-ton liquid scintillator, 9.24 kg of ^96^Zr can be loaded. The dispersion concentration of nanoparticles needs to be enhanced, because tens or hundreds of kilograms of candidate isotopes are required for the measurement of 0*νββ*. In addition, it is necessary to use raw materials enriched with ^96^Zr. If enriched raw materials of 90% ^96^Zris were used, the Zr dispersion concentration of 1 to 2 wt.% is required for loading the target amount of candidate isotopes. Therefore, further enhancement in the ZrO_2_ loading concentration is necessary.

When observing 0ubb of ^96^Zr, β-rays generated from the daughter nuclei of ^238^U and ^232^Th are expected to be a source of serious background. According to a previous report, the precursor ZrOCl_2_∙8H_2_O contains U and Th at several ppm [46]. Therefore, high-purity raw materials will be required. From ^238^U and ^232^Th, α-rays of 4.27 and 4.80 MeV are generated. Therefore, it is necessary to consider the influence of these α-rays on the background. According to a previous paper, the ratio of the scintillation efficiency of α-rays to β-rays (R(α/β)) of liquid scintillators is about 0.1 [47]. Hence, they would be observed in the energy spectrum at 0.42 and 0.48 MeVee for α-rays from ^238^U and ^232^Th, respectively. Hence, it is considered that the background of α rays near the Q value is limited. However, since the influence of β-rays generated from the daughter nuclei of ^238^U and ^232^Th cannot be ignored, it is necessary to purify the Zr raw material.

Furthermore, it is necessary to estimate the energy resolution. To estimate the energy resolution of these scintillators, it is necessary to measure with gamma rays with energy close to the *Q* value of 0*νββ*. However, since the liquid scintillator used in this study had a small volume, a gamma-ray photoelectric peak was not observed. Since it is necessary to increase the amount of liquid scintillator to observe the photoelectric peak, the immediate challenge is to establish the technology to construct a large-scale scintillator.

Finally, for double beta decay search, it is necessary to establish a technology for mass synthesis of nanoparticles. Since nanoparticles were synthesized in a batch process in this study, it is necessary to establish a flow-type continuous synthesis process in the future.

## Figures and Tables

**Figure 1 nanomaterials-11-01124-f001:**
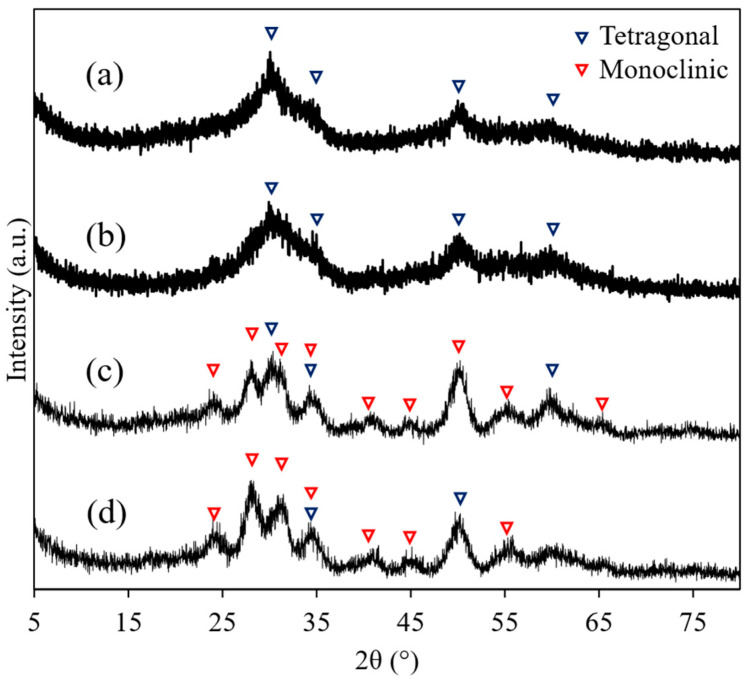
XRD patterns obtained for ZrO_2_ nanoparticles synthesized at various temperatures: (**a**) 250 °C, (**b**) 300 °C, (**c**) 350 °C, and (**d**) 400 °C.

**Figure 2 nanomaterials-11-01124-f002:**
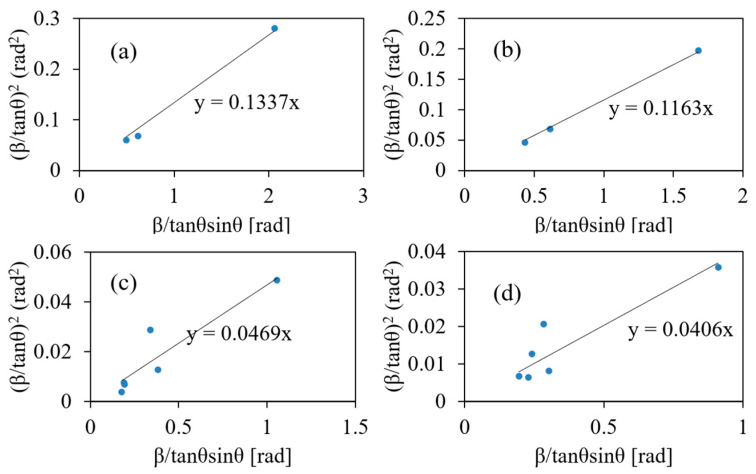
Halder–Wanger plot of the ZrO_2_ nanoparticles synthesized at (**a**) 250 °C, (**b**) 300 °C, (**c**) 350 °C, and (**d**) 400 °C obtained using Equation (2).

**Figure 3 nanomaterials-11-01124-f003:**
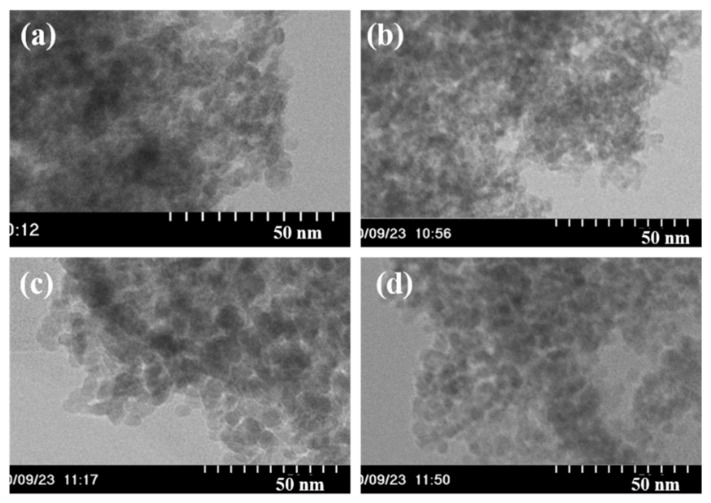
TEM images obtained for ZrO_2_ nanoparticles synthesized at various temperatures: (**a**) 250 °C, (**b**) 300 °C, (**c**) 350 °C, and (**d**) 400 °C.

**Figure 4 nanomaterials-11-01124-f004:**
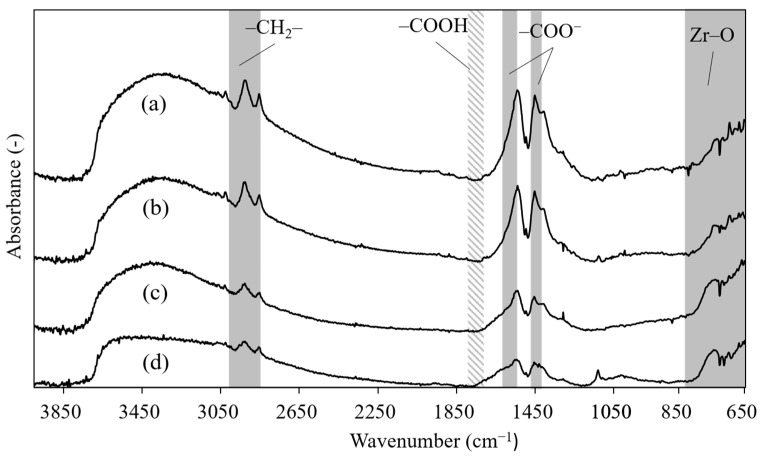
FTIR spectra obtained for PHA-modified ZrO_2_ nanoparticles synthesized at various temperatures: (**a**) 250 °C, (**b**) 300 °C, (**c**) 350 °C, and (**d**) 400 °C.

**Figure 5 nanomaterials-11-01124-f005:**
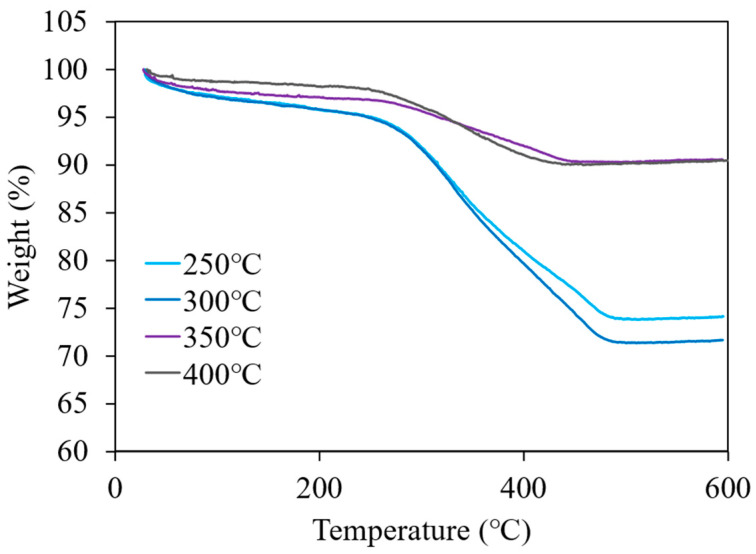
TGA curves obtained for PHA-modified nanoparticles synthesized at 250, 300, 350, and 400 °C under an air atmosphere and heating rate of 10 K/min.

**Figure 6 nanomaterials-11-01124-f006:**
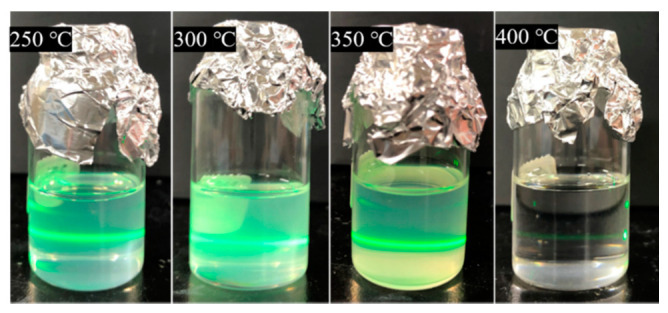
Photographic images of the PHA-modified ZrO_2_ nanoparticle dispersions in toluene synthesized at 250, 300, 350, and 400 °C.

**Figure 7 nanomaterials-11-01124-f007:**
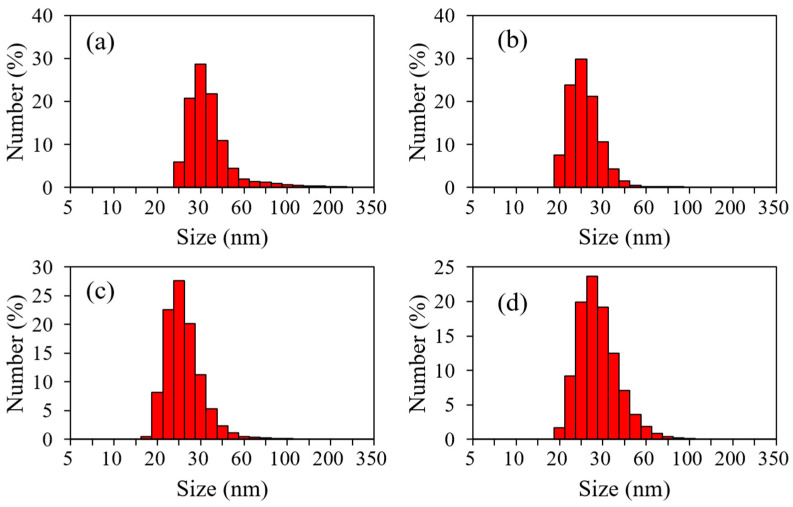
Number size distributions of the ZrO_2_ nanoparticles in the toluene dispersions synthesized at (**a**) 250 °C, (**b**) 300 °C, (**c**) 350 °C, and (**d**) 400 °C obtained using dynamic light scattering.

**Figure 8 nanomaterials-11-01124-f008:**
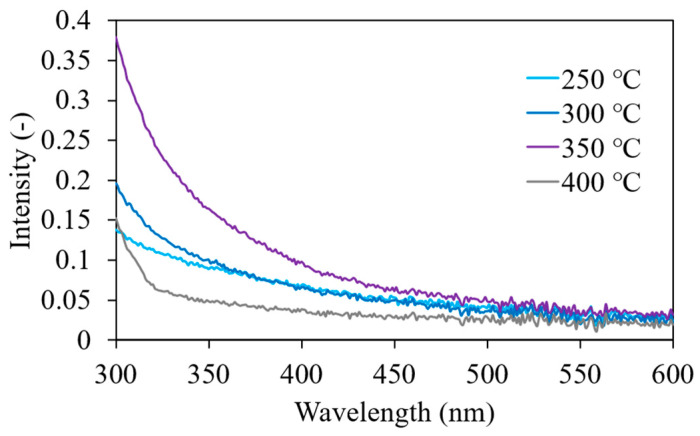
Absorption spectra obtained for the PHA-modified ZrO_2_ nanoparticle–toluene dispersion synthesized at 250, 300, 350, and 400 °C.

**Figure 9 nanomaterials-11-01124-f009:**
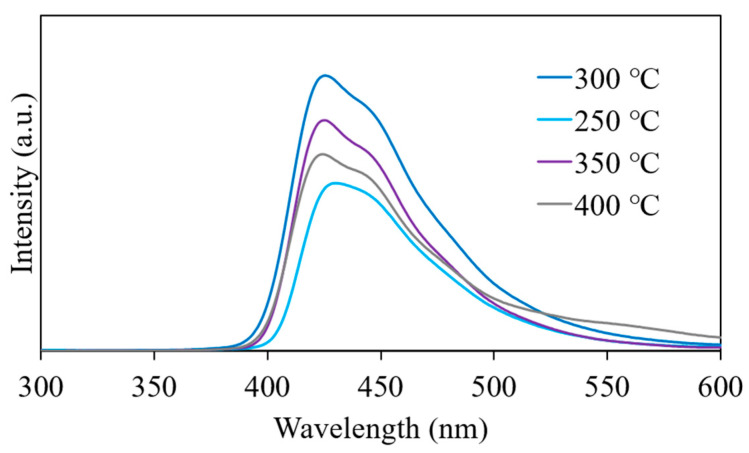
X-ray-induced radioluminescence spectra obtained for the liquid scintillator incorporating PHA-modified ZrO_2_ nanoparticles synthesized at 250, 300, 350, and 400 °C.

**Table 1 nanomaterials-11-01124-t001:** The surface PHA modification density of the ZrO_2_ nanoparticles and Zr concentrations of the nanoparticle dispersions.

Product	Weight Loss (%) from TGA Measurements	Modification Density (Molecules/nm^2^)	Zr Concentration (wt.%) from ICP-AES
250 °C	23.0 ± 0.1	2.8 ± 0.0	1.1 ± 0.1 × 10^−1^
300 °C	24.3 ± 1.3	3.1 ± 0.2	3.3 ± 0.4 × 10^−1^
350 °C	8.63 ± 1.67	0.9 ± 0.2	5.7 ± 0.3 × 10^−2^
400 °C	8.42 ± 0.24	0.9 ± 0.3	9.2 ± 0.1 × 10^−3^

## Data Availability

The data presented in this study are available within this article.

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
