# Peer review of "Fabrication of Liquid Scintillators Loaded with 6-Phenylhexanoic Acid-Modified ZrO2 Nanoparticles for Observation of Neutrinoless Double Beta Decay"

_nanomaterials, 2021, doi:10.3390/nano11051124_

Round 1
Reviewer 1 Report
In my opinion, the presented results are consistent and presented in a clear way. However, some minor issues could be improved:
Crystallite sizes revealed by XRD are only weakly consistent with observations using TEM. Moreover, no estimation of uncertainties is reported for the XRD. I am not against the general conclusion, as sometimes it is difficult to obtain similar results using so much different techniques, but big XRD error values could "bring" the results closer. If possible, this could be improved.
FT-IR figures have a horizontal scale ending somewhat unfortunately at 650 cm-1. If it was a device range effect it should be mentioned in the text. If not, I would gladly see a little bit more towards lower energies.
The absorbance measurement seems to be described in a rather sketchy way. Some details would be welcome.
Reviewer 2 Report
A novel technique to develop liquid scintillators to investigate neutrinoless double beta decay based on loading nanoparticles in the scintillator is presented.
The advantages and challenges of developing such a double beta decay detector are clearly posed in the introduction providing a great deal of relevant references.
Many different techniques have been applied to characterize the nanoparticles and the toluene-based-scintillator prepared.
The work done is thorough and it is clearly described; the conclusions drawn from each of the presented results are highlighted.
For all these reasons, I recommend the publication in Nanomaterials. I list below (indicating line in the manuscript) some suggestions and comments intended to help to complete or precise some aspects.
15: I suggest completing saying "nuclear and particle physics"
16-17: The relevance of double beta decay does not directly imply the necessity of developing loaded scintillators. I suggest rewrinting the sentence "Consequently, ..." as:
"The development of organic liquid scintillators with high transparency and a high concentration of the target isotope would be very useful for neutrinoless double beta decay experiments."
34: I propose changing "verification" to "identification"
35: I propose changing "discovered" to "confirmed"
38: I propose changing "presence of 0νββ affirms" to "occurrence of 0νββ would confirm"
41: to provide a reference illustrating the high expected half-lives of 0νββ you could give, instead of Ref. [3] which is focused on liquid scintillator experiments, a recent general review of experimental results like this one:
Neutrinoless Double-Beta Decay: Status and Prospects, Michelle J. Dolinski, Alan W. P. Poon, Werner Rodejohann, Annual Review of Nuclear and Particle Science, 69, 2019, 219-251
47: There have been recent and important results from double beta decay experiments like CUORE and CUPID (bolometers), MAJORANA and GERDA (Ge detectors) and KamLAND-Zen (scintillators), so I suggest updating references [4,5,6]. Alternatively, you could just provide a general reference for all of them, like the review article indicated in the previous comment.
53: I think it would be more adecuate to say "although limited can be sufficient" instead of "is sufficient".
98: Could you provide a reference for the Q value of 96Zr quoted? You could indicate that the Q value corresponds to the transition energy.
110: as Experimental is an adjective I propose to change the title to "Experiment" or "Experimental work"
111-130: the process of preparing the nanoparticles is described in detail, but it is not indicated why the particular volumes or times considered were selected. Could that selection affect the final results? Other conditions have been tried? A comment on this would be of interest.
113-115: I guess you use Zr with natural abundance of 96Zr (2.8%). It would be possible to use in the process enriched materials to increase the amount of 96Zr?
117: there is any particular reason to focus on the temperature range from 250 to 400ºC? If there are practical limitations to work in that range or if some improvements could be expected at other temperatures, you could comment on.
178: a brief explanation on how the given crystallite sizes were deduced would be useful.
188: you could precise how the reported uncertainty in the diameters is estimated.
Table 1: some uncertainty could be quantified for the values presented in the table? A comment on the order of the uncertainties would be useful at least, to better assess the behaviour with temperature of the different parameters.
265-266: the observed aggregation was expected? Is it undesirable?
294: you state that a similar emission is expected for the beta-ray-induced scintillation from 96Zr that for X-rays. Which was the energy range of the X-rays used? Electrons from the double beta decay would be at the MeV range. Do you plan a direct measurement of the scintillation using an electron source? If so, you could mention it.
307-308: I suggest saying better "synthesized to assess their suitability for use in double beta decay experiments."
General comments:
- it would be useful, if it is possible, in order to compare with other double beta decay searches, to quantify the amount of emitters (96Zr) that would be available in a particular volume of the whole scintillator detector.
- there are plans to carry out in the next future a double beta decay search based on the developed technique? If so, it would be worthwhile to comment on as outlook.
Reviewer 3 Report
The manuscript "Fabrication of liquid scintillators loaded with 6-phenylhexanoic acid-modified ZrO2 nanoparticles for observation of neutrinoless double beta decay" describes carefully the production technique of a new liquid scintillator for double beta decay observation. The possibility of use ZrO2 nanoparticles is the fact of use active target isotope directly diluted in the scintillator.
The nanoparticles
ZrO2 nanoparticles synthesis at different temperature was tested and the material characterised in terms of nanoparticle size, transparency and variation of density. In the conlusion the best temperature for the production of scintillator is at 300 °C.
The characterization is rigorous and well described but i think that an anlysis of possibile intrinsic background must be added. This because in the title the main goal of usage is explicitely the bb0n event that requires ultralow levels of background. So I strongly ask to add possible numbers on the level of impurities (e.g. U and Th) that can contaminate the material during the preparation.
Moreover, I ask the authors to implement these corrections:
line 40: change "difficult" with "challenging"
line 42: please add the formula for the experimental sensitivity (e.g. Equation 1 in Appl. Sci. 2021, 11, 1606. https:// doi.org/10.3390/app11041606 or Equation 14 in Annu. Rev. Nucl. Part. Sci. 2019. 69:219–51 ) and describe it. You see that in this formula are included mass and detector resolution but there is also the background level that should be minimized (so the general comment I asked before comes from this). Also the resolution study of the scintillation material should be mentioned somehow.
line 98 : change "background noise" with "g-ray environmental background". I think you should also consider the alpha background coming from natural chains of 238U and 232Th that release alpha up do few MeV. How these particles affect the spectra? (Maybe lower level of MeVee, so in any case below the roi?)
Section 2: a general comment is to insert a reference per each technique mentioned (subcritical hydrothermal method, XRD, TEM and so on)
line 116: please add why the pH to 5.8 is so important
Figure1: are the patterns in Figure 1 fitted with Equation (1)? If yes, I would suggest to add a figure with the fit results
line 178 and 188: i do not understand the difference between these vaules and why at 178 there are no uncertainty. moreover, what is the limit that allows me to define a single-nanometer sized crystallites?
Figure 3 and line 200: enlarge the figure and if you can extend the x axis in the way that the structure at 650-850 cm^-1 can be more evident. Moreover make the caption of Figure 3 more explicative highliting the attention on wanted and unwanted bands.
208: specify why it is important the reduction of COOH and CH2 (are they quenchers?)
line 225: Reading the Plot in Fig 4, I do not see that "Almost no weight change was observed at >500 °C for any of the 225 samples": at 250 and 300° the loss is of about 30% that for me is significative. So please modify the sentence or explain why you claim that for all the samples there is no weight change!!!
line 248: what was the starting point of the preparation? how many grams of crystals in how much of toluene? Write that it is described in section "Experimental"
line 251: The main goal on the Tyndall scattering measurement technique is the testing of transparency of the compound?
line 262: the estimated sizes of nanoparticles in the dispersion is evaluates as is the average of the distribution? if yes, please add uncertainty on it
Round 2
Reviewer 2 Report
I think all the raised issues have been properly addressed by authors and then I consider the manuscript is ready for publication.
I would like to thank the authors for the detailed explanations in their answer.